# Predictors and Consequences of Sac Shrinkage after Endovascular Infrarenal Aortic Aneurysm Repair

**DOI:** 10.3390/jcm11113232

**Published:** 2022-06-06

**Authors:** Sébastien Michel Vedani, Séverine Petitprez, Eva Weinz, Jean-Marc Corpataux, Sébastien Déglise, Céline Deslarzes-Dubuis, Elisabeth Côté, Jean-Baptiste Ricco, François Saucy

**Affiliations:** 1Department of Vascular Surgery, Centre Hospitalier Universitaire Vaudois, 1011 Lausanne, Switzerland; sebastien.vedani@unil.ch (S.M.V.); severine.petitprez@chuv.ch (S.P.); eva.weinz@unil.ch (E.W.); jean-marc.corpataux@chuv.ch (J.-M.C.); sebastien.deglise@chuv.ch (S.D.); celine.deslarzes@chuv.ch (C.D.-D.); elisabeth.cote@chuv.ch (E.C.); 2Department of Clinical Research, University of Poitiers, 86073 Poitiers, France; jean.baptiste.ricco@univ-poitiers.fr; 3Service of Vascular Surgery, Ensemble Hospitalier de la Côte, 1110 Morges, Switzerland

**Keywords:** abdominal aortic aneurysms, endovascular aneurysm repair (EVAR), Endurant, sac shrinkage, predictive factors

## Abstract

**Background:** Aneurysm shrinkage has been proposed as a marker of successful endovascular aneurysm repair (EVAR). We evaluated the impact of sac shrinkage on secondary interventions, on survival and its association with endoleaks, and on compliance with instructions for use (IFU). **Methods:** This observational retrospective study was conducted on all consecutive patients receiving EVAR for an infrarenal abdominal aortic aneurysm (AAA) using exclusively Endurant II/IIs endograft from 2014 to 2018. Sixty patients were entered in the study. Aneurysm sac shrinkage was defined as decrease ≥5 mm of the maximum aortic diameter. Univariate methods and Kaplan–Meier plots assessed the potential impact of shrinkage. **Results:** Twenty-six patients (43.3%) experienced shrinkage at one year, and thirty-four (56.7%) had no shrinkage. Shrinkage was not significantly associated with any demographics or morbidity, except hypertension (*p* = 0.01). No aneurysm characteristics were associated with shrinkage. Non-compliance with instructions for use (IFU) in 13 patients (21.6%) was not associated with shrinkage. Three years after EVAR, freedom from secondary intervention was 85 ± 2% for the entire series, 92.3 ± 5.0% for the shrinkage group and 83.3 ± 9% for the no-shrinkage group (Logrank: *p* = 0.49). Survival at 3 years was not significantly different between the two groups (85.9 ± 7.0% vs. 79.0 ± 9.0%, Logrank; *p* = 0.59). Strict compliance with IFU was associated with less reinterventions at 3 years (92.1 ± 5.9% vs. 73.8 ± 15%, Logrank: *p* = 0.03). Similarly, survival at 3 years did not significantly differ between strict compliance with IFU and non-compliance (81.8 ± 7.0% vs. 78.6 ± 13.0%, Logrank; *p* = 0.32). **Conclusion**: This study suggests that shrinkage ≥5 mm at 1-year is not significantly associated with a better survival rate or a lower risk of secondary intervention than no-shrinkage. In this series, the risk of secondary intervention regardless of shrinkage seems to be linked more to non-compliance with IFU. Considering the small number of patients, these results must be confirmed by extensive prospective studies.

## 1. Introduction

Since the introduction of endovascular aneurysm repair (EVAR) by Parodi, et al. in 1991 [1], EVAR has become the most-used method for abdominal aortic aneurysm (AAA) repair and, according to recent European guidelines, is now the preferred strategy whenever possible [2]. Although there is a clear early survival benefit to EVAR, when compared to open surgery [2], this early benefit weakens over time, due to rare secondary rupture, non-exceptional conversion to open repair and recurrent endovascular secondary procedures, as shown by randomized controlled trials and large cohort trials [3,4,5,6].

Sac shrinkage, defined as a reduction of ≥5 mm of the maximum sac diameter, is considered a reliable indicator of success [5,7], and is associated with better outcomes compared to stable or enlarging aneurysms [6,8,9]. However, predictors of sac behavior after EVAR, such as aneurysm neck length, angle and diameter, aneurysm sac diameter, thrombi and calcifications, as well as strict compliance with instructions for use (IFU), are still under discussion [10,11,12,13].

The aim of this study was to evaluate the association between sac shrinkage and secondary intervention, as well as survival following EVAR in a single center, taking into account compliance with IFU.

## 2. Materials and Methods

This observational retrospective study was conducted at the Centre Hospitalier Universitaire Vaudois (CHUV) in Lausanne, Switzerland, on all consecutive patients receiving EVAR for an infrarenal abdominal aortic aneurysm (AAA) from January 2014 to July 2018. The local ethics committee approved the study protocol (Project ID: 2018-01621). All selected EVAR procedures were performed in this series using Endurant II/IIs endograft (Medtronic, Minneapolis, MN, USA). 

### 2.1. Exclusion Criteria

We excluded from the study patients who refused to sign the consent form for analysis of their data, patients treated urgently, patients with previous aortic surgery, patients with a juxta- or suprarenal AAA, patients with a thoracoabdominal aneurysm or with an associated thoracic aneurysm and patients with an isolated iliac aneurysm or receiving an iliac branched device (Figure 1). All 60 patients included were followed for more than a year after EVAR with clinical consultations, but in some of them, control CTAs were done before the anniversary date. From the 129 initial patients receiving EVAR, 69 were excluded, leaving 60 patients for analysis. Patients treated with other endografts were also excluded.

### 2.2. Data Management

Data were retrieved from the vascular unit database, SecuTrial^®^ (interActive Systems GmbH, Berlin, Germany) and the CHUV electronic patient database. All data were anonymized. 

Data from CTAs were analyzed with Vue-PACS^®^ (Carestream Health, Ontario, ON, Canada). Preoperative CTAs were compared to postoperative CTAs, both with 1 mm slices and centerline measurements using outer-to-outer diameters. If more than one postoperative CTA was done, the second of the two scans was considered the index CTA compared to the preoperative CTA. 

Briefly, the following imaging data, taken from CTAs, were examined: neck diameter with four measurements starting from the level of the lowest renal artery with the calculation of the maximum diameter; neck length calculated as the distance between the lowest renal artery and the beginning of the aneurysm sac; degree of suprarenal neck angulation, calculated according to van Keulen, et al. [14] as the angle between the longitudinal axis of the suprarenal aorta and the longitudinal axis of the abdominal aortic aneurysm. In addition, thrombi and calcification at the aneurysm neck were measured according to wall extent. Thrombi or calcification present in one quadrant of the neck circumference was coded 25%, 50% in two quadrants, 75% in three quadrants and 100% in four quadrants. 

The maximum aneurysm sac diameter was measured using the centerline. Aneurysm sac calcification was measured, as neck calcification, as a percentage of the circumference covered. We measured the maximum aortic diameter (*MAD*) of the sac, and the size of the flow lumen maximum diameter (*FLMD*), to calculate the thrombus index (TI) of the sac, using the formula TI = [(*MAD − FLMD*)/*MAD*]. Patency of the inferior mesenteric artery (IMA), and the number of patent lumbar arteries within the sac, were also evaluated by CTA, together with the maximum diameter of each common iliac artery (CIA). According to Rouby, et al. [15], any CIA of a maximum diameter ≥17 mm was classified as an aneurysm. 

Finally, endoleaks were classified as Type I, Type II, or Type III. Compliance with the instructions for use (IFU) for Endurant^®^ endoprosthesis (https://www.medtronic.com/us-en/healthcare-professionals/products/cardiovascular/aortic-stent-grafts/endurantii/indications-safety-warnings.html, accessed on 27 May 2022) was analyzed, including neck length ≥10 mm, neck angulation ≤60 degrees, neck diameter 19–32 mm, and iliac diameter 8–25 mm. IFU were coded as a single binary variable. Any EVAR procedure with a single unmet IFU instruction was classified as non-compliant. Patient demographics; clinical features; and preoperative cardiological [16], respiratory [17] and renal [18] assessments were evaluated with ASA scoring, and entered into the database.

### 2.3. Endpoints

The primary efficacy outcomes were (1) a 1-year shrinkage rate and (2) an aneurysm-related reintervention rate. According to the Society of Vascular Surgery’s guidelines [5], sac shrinkage was defined as a decrease of ≥5 mm of the preoperative maximum AAA sac diameter. For analysis purposes, the cohort was divided into two groups: (A) sac shrinkage with a maximum diameter reduction of ≥5 mm) and (B) no shrinkage, defined as stabilization (maximum diameter reduction of <5 mm) or expansion (maximum diameter increase of >5 mm). The primary safety outcome was patient survival.

### 2.4. Statistical Analysis

Univariate analysis was performed using t-test or ANOVA for normally distributed continuous variables and Wilcoxon rank-sum test for non-normally distributed continuous variables. Fisher’s test (two-sided) was used for categorical variables. Patient survival, shrinkage during follow-up and freedom from reintervention were calculated by the Kaplan–Meier method. The Logrank test was used to compare groups of interest. A *p*-value <0.05 was used to assess statistical significance. All analyses were performed with SPSS V.28 (IBM Corp) and R version 4.1.1 (The R Foundation for Statistical Computing).

## 3. Results

### 3.1. Population Demographics and Baseline Characteristics

Among the 60 patients of a mean age of 75.3 ± 8.3 years included in the cohort, most were men (*n* = 49, 81.7%). Hypertension (*n* = 49, 81.7%) and history of smoking (*n* = 42, 70.0%) were the most frequent comorbidities (Table 1). General anesthesia was performed in 52 patients (86%), the median duration of the EVAR procedure was 80 min [interquartile range (IQR); 66–105 min], and median hospital stay was 6 days [IQR; 3–7 days]. The median follow-up was 26.6 months [IQR: 15.8–40.8]. Aneurysm characteristics—including maximum aneurysm diameter (59.9 ± 11.9 mm) and aneurysm neck diameter, length and angle, together with IMA and lumbar arteries patency, thrombus index of the sac and iliac artery diameter—are described in Table 1.

### 3.2. Sac Shrinkage

The median follow-up time for the two groups with and without shrinkage was similar (825 days vs. 906 days, *p* = 0.49). The median imaging follow-up time between EVAR and postoperative index CTA was identical for patients with and without shrinkage (322 days vs. 343 days, *p* = 0.40). At one year, sac Shrinkage ≥5 mm was observed in 26 patients (43.3%), and no shrinkage in 34 patients (56.7%). In the group without shrinkage, 33 patients were stable, and 1 patient had sac expansion (Table 2). Life table analysis with Kaplan–Meier plots for the whole series found a 1-year shrinkage rate of 36.0 ± 6.8% (Figure 2).

### 3.3. Demographics and Comorbidities

Shrinkage was associated with hypertension. Shrinkage occurred in 51.0% of patients with hypertension and 9.1% of patients without hypertension (Fisher’s exact test, *p* = 0.016). No other demographics and comorbidities were significantly associated with sac shrinkage (Table 2).

### 3.4. Aneurysm Morphology

A one-way, between-groups analysis of variance was conducted to explore the impact of neck length, neck diameter and neck angle on sac shrinkage (Table 2). There was no statistically significant difference at *p* < 0.05 level in AAA shrinkage for these variables. Similarly, AAA maximum diameter, AAA thrombus index, iliac aneurysm, IMA patency and number of patent lumbar arteries within the aneurysm were not significantly associated with sac shrinkage. Finally, non-compliance with instructions for use (IFU) in 13 patients (21.6%) was not significantly associated with sac shrinkage (Table 2).

### 3.5. Endoleaks

Endoleaks were observed in 17 (28.3%) patients: Type I (*n* = 4), Type II (*n* = 11), Type III (*n* = 2). Among these 17 patients, 4 presented sac shrinkage, and 13 had no shrinkage (Table 2). In other words, sac shrinkage was observed in 4 of 17 patients (23.5%) with an endoleak, and 22 of 43 patients (51.2%) without endoleak (*p* = 0.052). Therefore, we cannot rule out the null hypothesis. Furthermore, endoleaks should be considered as time-events and analyzed by Kaplan–Meier. At 1 year, the Kaplan–Meier plot for sac shrinkage was 46.1 ± 8.0% in patients without endoleak, and 13.0 ± 8.0% in patients with any type of endoleak (Logrank: *p* = 0.08) (Figure 3). However, given the small sample and the lack of power of the study, the non-significant association between endoleaks and sac shrinkage should be interpreted with caution.

### 3.6. Reintervention

Six patients (10%) underwent AAA-related secondary interventions, five for endoleaks (four Type I, one type II) and one for endograft occlusion. Among these 6 reinterventions, 3/26 (11.5%) were carried out in the shrinkage group and 3/34 (8.8%) in the no-shrinkage group (*p* = 0.73) (Table 2). Three years after index EVAR, with Kaplan–Meier plots, freedom from secondary intervention was 85 ± 2% for the entire series, 92.3 ± 5.0% for the shrinkage group and 83.3 ± 9% for the no-shrinkage group (Logrank test: *p* = 0.49) (Figure 4).

### 3.7. Instructions for Use

In this series, 13 patients (21.6%) received an Endurant II endograft outside the recommended IFU. Among those, eight patients had a neck angle >60°, two patients a neck length <10 mm, four patients a neck diameter >32 mm, and four patients an iliac aneurysm diameter >25 mm. Five patients had more than one instruction outside IFU. In this study, failure to comply with at least one IFU instruction had no significant impact on sac shrinkage at 1 year (Table 2).

However, as shown in Table 3, 4/12 patients (41.7%) outside IFU presented a type I/III endoleak, compared to 2/48 patients (4.2%) within IFU (*p* = 0.01). Similarly, 3/12 patients (25.0%) outside IFU had to undergo a secondary intervention compared to 3/48 (6.3%) patients within IFU (*p* = 0.05). A life table with Kaplan–Meier plots found, at 3 years, a freedom from the secondary intervention of 92.1 ± 5.9% in the compliant group vs. 73.8 ± 15% in the non-compliant group (Logrank: *p* = 0.03) (Figure 5). It should be noted that the higher risk of secondary intervention in the non-compliant group was not associated with a significant difference in sac shrinking (Table 2).

### 3.8. Survival

The overall mortality rate was 16.7% (10 patients); the median survival time after EVAR was 26 months. No death was related to EVAR but cardiovascular and respiratory diseases. In the whole series, survival at 3 years was 81.1 ± 6.0%. Survival at 3 years was not significantly different in the shrinkage group (85.9 ± 7.0%) compared to the no-shrinkage group (79.0 ± 9.0%, Logrank; *p* = 0.59) (Figure 6). Similarly, survival at 3 years in patients within IFU (81.8 ± 7.0%) was not significantly different from that of patients outside IFU (78.6 ± 13.0%, Logrank; *p* = 0.32) (Figure 7).

## 4. Discussion

The present study confirms that sac shrinkage occurred in almost half of the patients, the typical rate found in other studies. Only one patient (1.6%) from our cohort had sac expansion, a lower rate than the commonly reported studies [9,11,19,20]. In our study, sac shrinkage ≥5 mm of maximum diameter was not associated with a significant increase of secondary interventions or mortality. Indeed, many authors have reported sac shrinkage as a reliable indicator of successful treatment with reduction of aneurysm-related mortality, as well as improvement of freedom from secondary reintervention [6,7,21,22]. In this series, we observed no aneurysm-related death, and comparable 3-year survival rates of 85.9% and 79.0% in patients with and without shrinkage (*p* = 0.59). In other series, mortality plots for shrinkage and no shrinkage groups appear diverged after the 3-year follow-up. Our follow-up being limited to 3 years may explain this difference. 

In this study, contrary to other series [11,21], morphological factors such as neck length, angle and diameter were not predictors of sac shrinkage, likewise for AAA maximum diameter, iliac aneurysms, and IMA or lumbar arteries patency. Preoperative AAA maximum diameter and its relation to sac evolution are still debated. Some studies have shown large AAA diameter as a significant predictor of sac regression [7,23]. Other studies found that aneurysms of small diameter regress more and have better outcomes [24]. Whilst our analysis fails to show significant association between morphological factors including neck length, angle and diameter and shrinkage, presumably due to the relatively small numbers, it does show that strict adherence to all IFU instructions was significantly associated with sac shrinkage.

The thrombotic burden of the aneurysm has also been studied for sac shrinkage, but continues to be the subject of debate. Sadek, et al. [25] found that highly thrombosed aneurysms, assessed by volume measurement, regress more than others, possibly by reducing re-entry from lumbar arteries and IMA. In contrast, other authors have shown that aneurysms with a low thrombotic burden regress more than those with a large thrombus [26]. In our study, the thrombotic burden within the sac, assessed by a ratio between maximum sac diameter and maximum lumen diameter, was not significantly correlated with sac shrinkage. Some studies have shown in non-treated AAA that the thrombus provokes inflammation which may inhibit remodeling and thus sac shrinkage [27]. 

Comparison of these studies on the thrombus burden is difficult due to the varied technologies used by the authors. Our definition of the thrombus burden differs from those published, but it has the advantage of being a simple and easily reproducible measurement that does not need complex volume-measuring technology. However, we agree that this method does not take into account the tridimensional aspect of thrombi. Specific studies are required to define better the potential influence of thrombi on shrinking after EVAR.

Another potential predictor of sac shrinkage is aneurysm sac calcification. We did not find any correlation between AAA calcification and sac regression. An inverse correlation between calcium build-up and the expansion rate of AAA has been reported by Nakamaya, et al., in a cohort of 414 patients [28]. On the other hand, Lindholt, et al. [29] have shown in a cohort of 122 patients that AAA calcification >50% was associated with a higher rate of sac regression. One possible explanation is the interaction between calcification and smooth muscle cells. Further work to establish whether AAA calcification plays any role regarding sac shrinkage is required.

Some studies have shown that shrinkage was more frequently seen in young patients (<75 years) [11,21,26]. We did not find such an association. In our study, the mean age was 76.9 ± 7.0 years in patients without shrinkage compared to 73.2 ± 9.0 years in patients with shrinkage (*p* = 0.08). This trend is not significant, but considering our study’s relatively small number of patients, we cannot exclude a lack of statistical power.

However, our study confirmed some statements found in the literature. Hypertension was significantly correlated with regression (*p* = 0.016), as reported by O’Donnell [9]. Even if solid evidence is still missing, this inverse correlation may be explained by antihypertensive drugs, especially calcium channel blockers, which could induce sac regression through a possible down-regulation of the inflammatory process [30]. 

Statin therapy and its relation to AAA has also been widely studied. We could not show any correlation between statin therapy and sac shrinkage. A meta-analysis showed no significant reduction in AAA growth for patients under statins, although it seemed to improve outcomes after EVAR [31]. The supported mechanism of sac shrinkage in patients receiving statin therapy is a decrease of matrix metal proteases (MMP) in the arterial wall [32]. Still, the role of statin therapy regarding sac shrinkage following EVAR remains debated. 

In this series, all EVAR procedures were performed with Endurant II/IIs endograft, the most-used endograft in our center. As shown in some studies [33,34,35], sac shrinkage differs according to different types of endografts. This is mainly explained by the specificity of each endograft, regarding anatomy, with limitations reported in IFU. 

In our cohort, 13 patients (21.6%) were treated outside IFU, mainly neck angulation. Even if no significant difference in sac shrinkage was found following EVAR procedures, freedom from secondary intervention at 3 years was significantly higher when adherence to all IFU instructions was fulfilled (92.1 ± 5.9%), as opposed to cases where at least one instruction was not followed (73.8 ± 15.0%, *p* = 0.03). However, shrinkage and survival at 3 years were not significantly different between these two groups. Schanzer, et al. [11] found a correlation between EVAR outside IFU, secondary intervention and sac expansion. Our study confirms that strict adherence to IFU with Endurant II endograft reduced the risk of secondary intervention. Still, we did not find any correlation between sac shrinkage and strict adherence to IFU. Of note, and even allowing for strict IFU observance, some authors [36] noted no difference in patients treated within or outside IFU, regarding endoleak, reintervention and overall survival.

In this series, the presence of endoleak of any type (*n* = 17) was not significantly correlated with failure of sac shrinkage (38.2% vs. 15.4%, Fisher’s exact test 2 sided, *p* = 0.08). This negative result was confirmed by Kaplan–Meier plots (Logrank: *p* = 0.08). However, many studies have shown such an association [8,9,26], and the small number of patients leading to a loss of statistical power may explain our results, which differ from those in the literature.

In this study, the patency of the IMA, whatever its diameter, was not significantly associated with sac expansion. Today, the role of IMA in sac shrinkage remains controversial. Some authors have recommended preoperative IMA embolization to promote shrinkage [37,38]. To date, no study has shown that IMA patency is linked to the absence of sac shrinkage, and the benefice of preoperative embolization remains unproven. 

### Limitations of the Study

This study has several limitations. Firstly, it is a retrospective study from a single center. The small number of patients included in this observational study, with short follow-up, probably underestimated the importance of the correlation between non-compliance with IFUs and the absence of AAA shrinkage. Nevertheless, despite these limitations, we demonstrated the importance of compliance with IFU specific to this endoprosthesis, but due to the observational nature of the study, we could not prove a causal relationship. Moreover, as this study pertains only to Endurant ^®^ endografts, our results are only applicable to this brand of endografts.

## 5. Conclusions

This study suggests that shrinkage ≥5 mm at 1-year is frequent but not significantly associated with a better survival rate or a lower risk of secondary intervention compared with no shrinkage. In this series, the risk of secondary intervention regardless of shrinkage seems to be linked more to non-compliance with IFU. Considering the small number of patients, these results will need to be confirmed by large prospective studies.

## Figures and Tables

**Figure 1 jcm-11-03232-f001:**
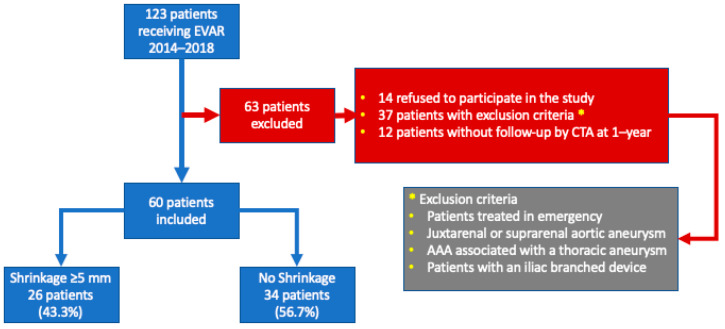
Flowchart of the study.

**Figure 2 jcm-11-03232-f002:**
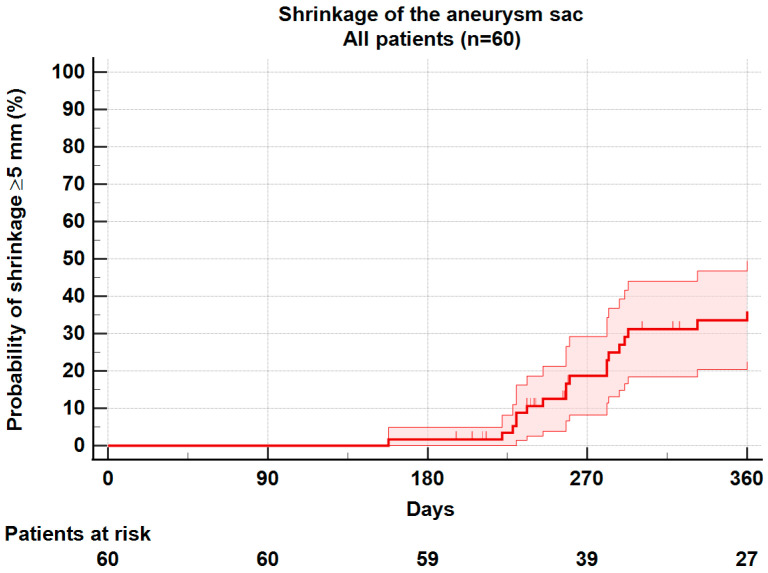
Kaplan–Meier plot for occurrence of sac shrinkage. Shrinkage at 1 year was 36.0 ± 6.8% for the whole series.

**Figure 3 jcm-11-03232-f003:**
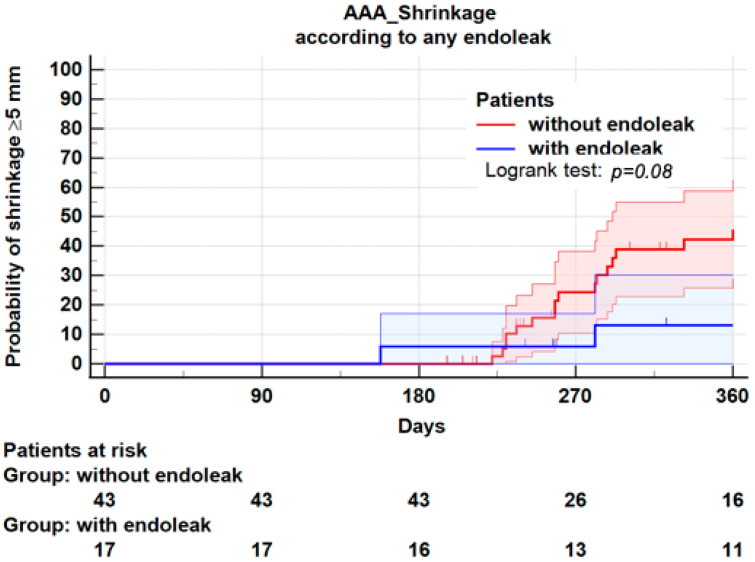
Kaplan–Meier plot for occurrence of sac shrinkage according to endoleak. Shrinkage at 1 year was 46.1 ± 8.0% in patients without endoleak, and 13.0 ± 8.0% in patients with any type of endoleak, *p* = 0.08.

**Figure 4 jcm-11-03232-f004:**
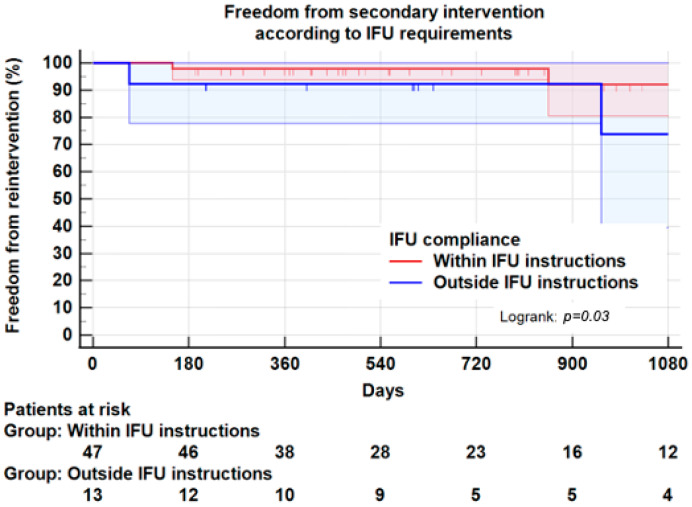
Kaplan–Meier plot for freedom from secondary intervention according to compliance with IFU instructions. At 3 years, freedom from secondary intervention was 92.1 ± 5.9% in patients compliant with IFU instructions, and 73.8 ± 15% in patients non-compliant with IFU instructions, *p* = 0.03.

**Figure 5 jcm-11-03232-f005:**
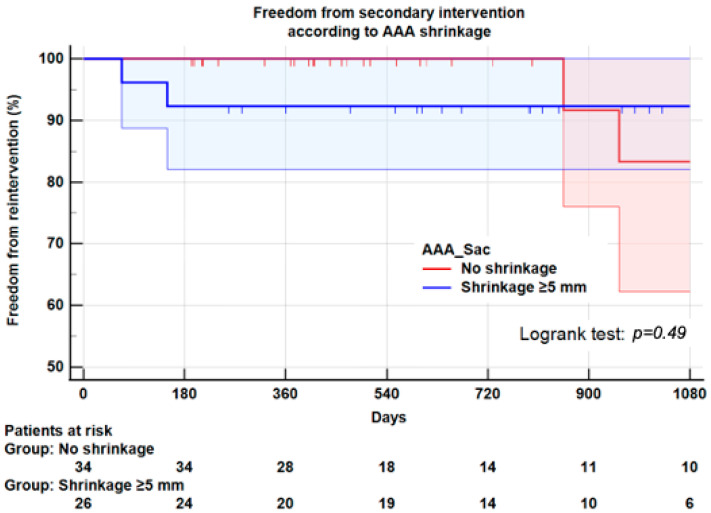
Kaplan–Meier plot for freedom from secondary intervention according to sac shrinkage. At 3 years, freedom from secondary intervention was 92.3 ± 5.0% in patients with sac shrinkage and 83.3 ± 9.0% in patients with no sac shrinkage, *p* = 0.49.

**Figure 6 jcm-11-03232-f006:**
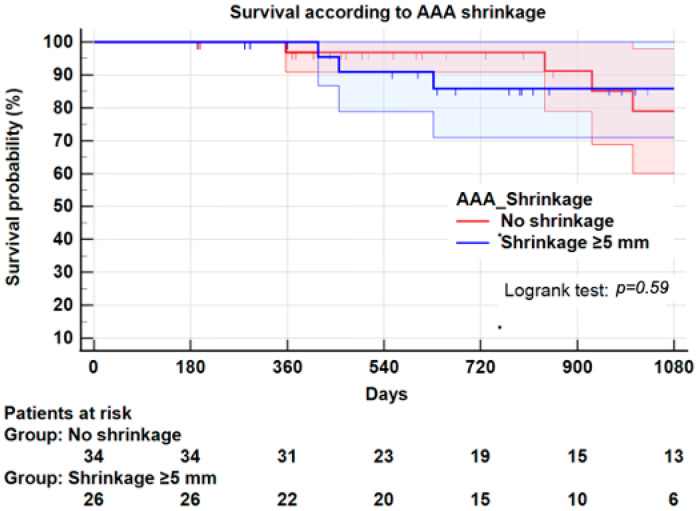
Kaplan–Meier plot for survival according to sac shrinkage. At 3 years, survival was 85.9 ± 7.0% in patients with sac shrinkage and 79.0 ± 9.0% in patients without sac shrinkage, *p* = 0.59.

**Figure 7 jcm-11-03232-f007:**
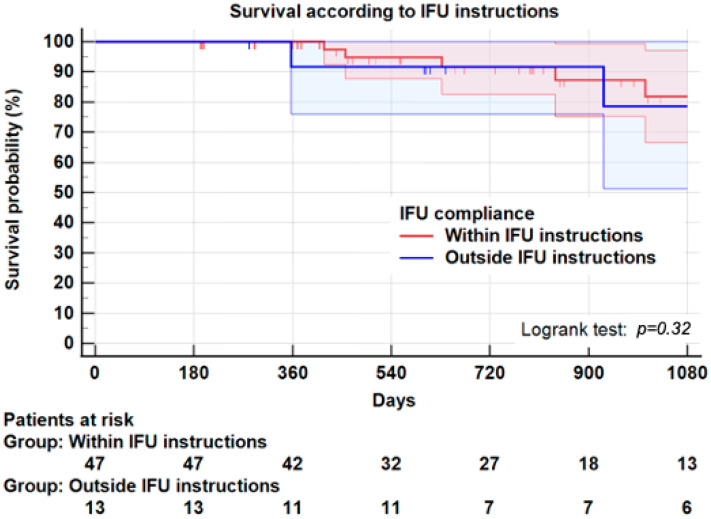
Kaplan–Meier plot for survival according to compliance with IFU instructions. At 3 years, survival was 81.8 ± 7.0% in patients compliant with IFU instructions and 78.6 ± 13.0% in patients non-compliant with IFU instructions, *p* = 0.32.

**Table 1 jcm-11-03232-t001:** Main characteristics of 60 patients undergoing EVAR for infrarenal aortic aneurysm.

Patient Characteristics		
**Mean Age, years (±SD)**	75.3 (±8.3)	
	**N**	**%**
**ASA score**		
**ASA 2**	16	26.7
**ASA 3**	42	70.0
**ASA 4**	2	3.3
**Male gender**	49	81.7
**Smoking history**	42	70.0
**Cardiac disease**	28	46.7
**Hypertension**	49	81.7
**Diabetes mellitus**	9	15.0
**Dyslipidaemia**	28	46.7
**Chronic renal insufficiency ***	13	21.7
**COPD**	20	33.3
**Antiplatelet therapy**	28	46.7
**Statins**	35	58.3
**Aneurysm Characteristics**	**Median**	**IQR**
**Aneurysm maximum diameter (mm)**	57	(54–61)
**Neck length (mm)**	28	(17–36)
**Neck maximum diameter (mm)**	25	(24–28)
**Neck angle (degree)**	27	(21–46)
**Iliac arteries maximum diameter (mm)**	18	(15–20)
**Thrombus Index (TI) of the sac ^†^**	0.5	(0–0.5)
**Other Characteristics**	**N**	**% or IQR**
**Inferior Mesenteric Artery patency**	45/60	75.0%
**Number of patent lumbar arteries (±SD)**	5	(4–7)

ASA: American Society of Anesthesiologists, * Chronic renal insufficiency was defined as GFR <60 mL/min/1.73 m^2^. ^†^ Thrombus index (TI) of the sac was calculated using the formula TI = *[(Maximum Aortic Diameter − Maximum Diameter of Flow Lumen)/Maximum Aortic Diameter]*. IQR: Interquartile range.

**Table 2 jcm-11-03232-t002:** Main characteristics of 60 patients undergoing EVAR for infrarenal aortic aneurysm according to sac shrinkage ≥5 mm.

Patient Characteristics			
	No Shrinkage N = 34	Shrinkage N = 26	*p*-Value
**Mean Age, years** (**±SD**)	76.9 ± 7	73.2 ± 9	0.08
	**N (%)**	**N (%)**	
**Male gender**	26 (76.5)	23 (88.5)	0.32
**Smoking history**	24 (70.6)	18 (69.2)	0.90
**Hypertension**	24 (70.6)	25 (96.2)	0.01
**Diabetes mellitus**	4 (11.8)	5 (19.2)	0.48
**Dyslipidaemia**	13 (38.2)	15 (57.7)	0.19
**Chronic renal insufficiency ***	8 (23.5)	5 (19.2)	0.76
**COPD**	12 (35.3)	8 (30.8)	0.78
**Antiaggregant therapy**	17 (50.0)	11 (42.3)	0.61
**Statins**	19 (55.9)	16 (61.5)	0.79
**Aneurysm Characteristics**	**Median (IQR)**	**Median (IQR)**	
**Aneurysm maximum diameter** (**mm**)	56 (52–60)	58 (55–62)	0.14
**Neck length** (**mm**)	29 (17–36)	25 (16–35)	0.55
**Neck maximum diameter** (**mm**)	26 (24–29)	25(22–28)	0.41
**Neck angle** (**degree**)	28 (23–46)	25 (16–45)	0.53
**Neck thrombus** (**% of circumference**)	25 (0–50)	0 (0–43)	0.21
**Thrombus Index** (**TI**) **of the sac ^†^**	0.50 (0–0.50)	0.50 (0–0.50)	0.78
**Iliac arteries maximum diameter** (**mm**)	18 (16–20)	17 (14–20)	0.26
**Number of patent lumbar arteries** (**±SD**)	5 (4–6)	6 (4–7)	0.96
**Other Characteristics**	**N (%)**	**N (%)**	
**Inferior Mesenteric Artery patency**	26 (76.5)	19 (73.1)	0.77
**Failure to comply with at least one IFU instruction** (***n* = 13 patients**) **^††^**	7 (20.6)	6 (23.1)	0.89
**IFU Neck length <10 mm**	1 (2.9)	1 (3.8)	0.85
**IFU Neck diameter > 32 mm**	2 (5.9)	2 (7.7)	0.78
**IFU Neck angle > 60 degrees**	4 (11.8)	4 (15.4)	0.72
**IFU Iliac diameter > 25 mm**	2 (5.9)	2 (7.7)	0.78
**Endoleaks**			
**All endoleaks**	13 (38.2)	4 (15.4)	0.08
**Endoleaks Types I–III**	4 (11.8)	2 (7.7)	0.60
**Endoleaks Type II**	9 (26.5)	2 (7.7)	0.08
**Reinterventions**			
**All Ever secondary interventions**	3 (8.8)	3 (11.5)	0.73

* Chronic renal insufficiency was defined as GFR <60 mL/min/1.73 m^2^. ^†^ Thrombus index (TI) of the sac was calculated as TI = *[(Maximum Aortic Diameter − Maximum Diameter of Flow Lumen)/Maximum Aortic Diameter]*, SD: Standard Deviation. ^††^ Compliance with all the instructions for use (IFU) for Endurant^®^ endoprosthesis, including neck length ≥10 mm, neck angulation ≤60 degrees, neck diameter 19/32 mm, and iliac diameter 8/25 mm. Failure to comply with at least one IFU instruction was found in 13 patients. Failure to comply with more than one IFU instruction was found in two patients. The Mann–Whitney U test was used to calculate the *p*-values for aneurysm characteristics, as the Shapiro–Wilk test suggests a violation of the assumption of normality for most of these variables. IQR: Interquartile range.

**Table 3 jcm-11-03232-t003:** Data concerning endoleaks and secondary interventions based on compliance with instructions for use (IFU) in 60 patients undergoing EVAR for infrarenal aortic aneurysm.

	Index EVAR within IFU N = 48	Index EVAR outside IFU N = 13	*p*-Value
Characteristics	N (%)	N (%)	
**All endoleaks**	12 (25.0)	5 (41.7)	0.25
**Endoleaks Type I–III**	2 (4.2)	4 (33.3)	**0.01**
**Endoleaks Type II**	10 (20.8)	1 (8.3)	0.32
**Reintervention ***	3 (6.3)	3 (25.0)	**0.05**

* All secondary interventions related to index EVAR procedure.

## Data Availability

Not applicable.

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
