# Peer review of "Predictors and Consequences of Sac Shrinkage after Endovascular Infrarenal Aortic Aneurysm Repair"

_jcm, 2022, doi:10.3390/jcm11113232_

Round 1

Reviewer 1 Report

The authors addressed most of the concerns of the original review and included discussion about differences in the findings in this study compared to prior studies, a lot of which is related to the small N of this study.

A few specific comments remain:

Your overall follow up time was 3 years.  Did you have further aneurysm sac behavior that you could report?  Did those who had shrunk >5mm continue to shrink?  Did others that did not shrink subsequently shrink or demonstrate growth?

On page 11, first paragraph, the authors discuss their finding of the lack of influence of mural thrombus on AAA sac behavior and that their method of quantifying thrombus varies from the other publications which use volumetric measures.  They discuss the ease of their approach but don’t discuss the shortcomings of their approach (single measure, not representative of the thrombus throughout the sac, etc).  The limitations of their method of quantification should be discussed

Figure 6  legend – both reported survival risk for “patients with sac shrinkage”.  One should be for patients without shrinkage.  Please correct.

Reviewer 2 Report

The manuscript of Vedani and coworkers examines the association of AAA shrinking with clinical and surgical variables in patients after EVAR. The results presented by the Authors identify in hypertension the only relevant variable associated with AAA shrinking. No association has been identified between shrinking (or IFU) and survival.

The relatively small group of during and short follow-up period might contribute to the paucity of results, as admitted from the Authors. Nevertheless, looking through the figures, it appears that the analysis suffers from disappointing inaccuracies.

The first limitation is on data description: it is not clear the distribution of AAA characteristics in table 1. Should we assume that all parameters were normally distributed (with some aortic diameters < 48mm)? Was, moreover, the type of analysis consistent with data distribution? It would be more informative to report median and InterQuartile Ranges and use only one type of test (namely non parametric Wilcoxon).

On table 2, there are some other points needing attention. The number of patent lumbar arteries needs to be better described, since reporting 4.8 or 5.4 patent arteries does not make sense. Here too, median and IQR values are a better choice.

The most disappointing evidence is that chi square test P values look wrong when comparing the prevalence of endoleaks in shrinking and not shrinking AAAs. To this reviewer, the figures of P values are lower and marginally significant for all (0.052) and type 2 (0.062) endoleaks. This result in such a small sample does not allow to rule out endoleaks as a possible cause of not shrinkage of an aneurysm.

Duration of follow-up is another critical issue. The Authors state in line 75 that “All 60 patients included were followed for more than a year after EVAR with clinical consultations, but in some of them, control CTAs were done before the anniversary date.” It is surprising that no CTAs were performed after 365 days and, looking at the figures, it appears that events were still recorded at the very end of follow up period. This reviewer wonders if a lengthier observation could have changed the results.

Minor points are:

The abstract is far too long, for the results presented;

On table 1, the term antiaggregant should be replaced.

 “The overall mortality rate was 16.7% (10 patients), median survival time after EVAR was 26 months.” Median survival time refers to the whole population or only to deceased patients?

The sentence “Unlike other studies, sac shrinkage ≥5 mm of maximum diameter was not associated with a significant increase of secondary interventions or mortality.” (line 267) needs to be rephrased, since it gives the impression that other studies identified shrinkage associated with worse prognosis.

The sentence “Hypertension was significantly correlated with regression (p= .047)” reports a P value not consistent with that on table 2;

Descriptions of Figure 4 and 5 are swapped.

One or two mistakes are on line 350 (“Frow now on”). “To date” might be a better choice.

Round 2

Reviewer 2 Report

A last remark is on the presentation of non-normally distributed values, since the actual presentation is not more informative than the previous version.

In literature, as index of data distribution, median is usually followed by IQR, which is the values of 25th and 75th percentile. Therefore, e.g., a skewed distribution of a variable could be described as "15 (13-28)". This allows to see that one quart of observations are grouped in a narrow range (13 to 15) while data are less "dense" in the range 15 to 28. This is essential in non-Gaussian distribution. Please modify the values accordingly.    

Author Response

Dear Reviewer,

That is correct and we apologize for the lack of comprehensibility.

The two tables (Table 1 and Table 2) have been modified and for variables, you'll now find the datas for 25th and 75th quartiles, as suggested. 

We hope this makes the results more understandable and we thank you for the reviews.

PS: the edited manuscript is now uploaded.

Vedani and colleagues